# Sleep Disorders in Low-Risk Preterm Infants and Toddlers

**DOI:** 10.3390/jpm13071091

**Published:** 2023-07-02

**Authors:** Domenico M. Romeo, Chiara Arpaia, Maria Rosaria Lala, Giorgia Cordaro, Francesca Gallini, Giovanni Vento, Eugenio Mercuri, Antonio Chiaretti

**Affiliations:** 1Pediatric Neurology Unit, Fondazione Policlinico Universitario A. Gemelli, IRCCS, L.argo A. Gemelli, 00168 Rome, Italy; domenicomarco.romeo@policlinicogemelli.it (D.M.R.); arpaia.chiara@gmail.com (C.A.); mariarosarialala.mrl@gmail.com (M.R.L.); cordarogiorgia@gmail.com (G.C.); eumercuri@gmail.com (E.M.); 2Pediatric Neurology Unit, Università Cattolica del Sacro Cuore, 00168 Rome, Italy; 3Neonatal Intensive Care Unit, Fondazione Policlinico Universitario A. Gemelli, IRCCS, L.go A. Gemelli 8, 00168 Rome, Italy; francesca.gallini@policlinicogemelli.it (F.G.); giovanni.vento@policlinicogemelli.it (G.V.); 4Neonatal Intensive Care Unit, Università Cattolica del Sacro Cuore, 00168 Rome, Italy; 5Department of Pediatrics, Università Cattolica del Sacro Cuore, 00168 Rome, Italy; 6Department of Pediatrics, Fondazione Policlinico Universitario A. Gemelli, IRCCS, Largo A. Gemelli, 00168 Rome, Italy

**Keywords:** sleep disorders, preterm, infants, toddlers, SDSC

## Abstract

Sleep disorders are particularly important in the development of children, affecting the emotional, behavioural, and cognitive spheres. The incidence of these disorders has been assessed in different types of populations, including patients with a history of premature birth, who, from the literature data, would seem to have an increased incidence of sleep disorders at school age. The aims of the present study are: (i.) to assess the presence of sleep disorders in a population of very preterm infants at 6–36 months who are at low risk of neurological impairments using the Italian version of the Sleep Disturbance Scale for Children (SDSC) adapted for this age group, and (ii.) to identify possible differences from a control group of term-born infants. A total of 217 low-risk preterm and 129 typically developing infants and toddlers were included in the study. We found no differences in the SDSC total and the factor scores between these two populations of infants. Low-risk preterm infants and toddlers showed similar incidences of sleep disorders to their term-born peers. Further clinical assessments will be needed to confirm these data at school age.

## 1. Introduction

Sleep problems are quite frequent in childhood, with a prevalence of 10–30% [1,2,3] especially related to bedtime resistance, delayed sleep onset, and night waking, which have a possible influence on children’s’ behavioural, emotional, cognitive and learning functioning, and, also, on the well-being of the whole family.

Several studies have also assessed sleep in children with developmental disabilities [4,5,6,7,8,9,10,11], but only a few of these studies focused on preterm children, especially at younger ages [12,13,14,15,16].

The majority of the studies on preterm school age children at both low and high risk of the sleep problems have shown that the prevalence of these problems increases with growing neurodevelopmental disabilities and with the decrease in the gestational age [12,13,16]. However, during childhood, preterm children reported more sleep problems than their typically developed peers, even in the absence of neurodevelopmental disabilities [14,15]. The findings regarding the incidence and the patterns of sleep problems in the preterm population are controversial and the results are discordant, mainly due to different inclusion criteria regarding gestational age and the age of assessment or presence of brain lesions, epilepsy, or other neurodevelopmental disabilities [12,13,14,15,16,17]. Furthermore, most studies assessed sleep disorders by using a sleep questionnaire not standardized for children.

The questionnaire itself could clearly be a faster and immediate method of data collection, although it has the limitation of not being a fully structured or objective method of evaluation.

On the other hand, there are some more objective methods, such as nocturnal polysomnography, that can provide more objective data, although they require a longer timeframe and a specific setting for administration [18].

Only a few studies have focused on preterm infants at pre-school age using standardized sleep questionnaires that are specifically validated for this age. Caravale et al. [14] recently explored the sleep patterns in preschool preterm children at the age of 2 years. The authors reported more medically-related sleeping problems in the preterm group than in the term one, such as nocturnal movement, restlessness during the night, and breathing problems. Nevertheless, they reported no differences between the preterm and term-born population on sleep patterns at bedtime, at rise time, and at nocturnal/daytime sleep durations. 

Another recent study [16] detected sleep disturbances in low-risk preterm children aged from 3 to 6 years old. No significant differences were reported according to the age of assessment or the gestational age. However, low-risk very preterm children reported a higher incidence of sleep disorders than their term-born peers, with higher scores in specific sleep problems, such as sleep-disordered breathing, sleep hyperhidrosis, and difficulty in initiating and maintaining sleep. 

In both of these studies, the authors used the Sleep Disturbance Scale for Children (SDSC) [19], one of the most widely used questionnaires for sleep disorders in paediatric patients with an appropriate psychometric criteria, and this scale has also been used in different cohorts of both high-risk and low-risk children for neurodevelopmental disorders [6,8,10,14,16]. The SDSC was developed for children from 6-years-old onwards; however, more recently, it was modified, validated, and standardized in a population of pre-school age children (3–6 years) and infants and toddlers (6–36 months) [20,21].

The possibility of the early detection of sleep problems in preterm children could be helpful for both families and physicians in order to promote strategies (such as parent-infant bonding, sensory stimulation, and the use of private family rooms) for mitigating postnatal stress in preterm infants [1,2,3,4,5,6,7,8,9,10,11,12].

The aims of the present study are: (i.) to assess the presence of sleep disorders in a population of very preterm infants at 6–36 months at low risk for neurological impairments, using the SDSC adapted for this age group, and (ii.) to identify possible differences from a control group of term-born infants.

## 2. Materials and Methods

### 2.1. Study Population

The infants included in the present study were part of a prospective project on preterm infants regularly followed at the Pediatric Neurology Unit and the Neonatal Intensive Care Unit of the Fondazione Policlinico Universitario A. Gemelli of Rome between January 2019 and December 2021. We included infants with a GA < 34 weeks gestation with no history of major prenatal, perinatal, or postnatal medical complications, and we examined them between the ages of 6 months and 36 months of corrected age. The exclusion criteria were the small size for gestational age (i.e., a weight below the 10th percentile), major cerebral lesions observed at US scans, congenital malformations, severe postnatal infectious diseases, metabolic complications, cerebral palsy, and epilepsy.

### 2.2. Sleep Assessment

Sleep disturbances were assessed using the SDSC validated for infants and toddlers at 6–36 months [20]. This questionnaire investigates the occurrence of sleep disorders during the previous 6 months, and it contains 19 items in a Likert-type scale with values 1–5 (higher numerical values reflect a higher clinical severity of symptoms). The total score consists of the sum of the 19 items retained, with a possible range from 19 to 95. A T-score of more than 70 (>95th centile) was considered “pathological”, while a T-score between 61 and 70 was considered a “suspect/borderline” score.

This questionnaire analyses six sleep disturbance factors representing the most common areas of sleep disorders in infants and toddlers: disorders in initiating sleep (DIS) related to sleep latency and problems in falling asleep; difficult in maintaining sleep (DMS) related to sleep duration, night awakenings, difficulty falling asleep again after waking, and nocturnal hyperkinesia; sleep-disordered breathing (SBD); parasomnias (PAR), which includes nightmares, sleepwalking, and sleep terrors; disorders of excessive somnolence factor (DOES) related to daytime somnolence and nonrestorative sleep; and sleep hyperhidrosis (SHY), which involves falling asleep sweating and night sweating.

This questionnaire was distributed to the primary caregiver of the infants, who was the mother in all of the cases, during the routine neurological assessment in our Unit.

The SDSC was further distributed to the primary caregivers of a group of children recruited via nurseries and considered as a control group. Questionnaires were filled out by the mothers during school hours under the supervision of the researchers who distributed the questionnaires, and no missing values were reported. Infants with mental, developmental, or physical disabilities, as well as those receiving ongoing prescription medication (i.e., antiepileptic drugs, antihistaminic drugs, benzodiazepine, melatonin, etc.) were all excluded, given that these aspects could all negatively influence the development of sleep and could, therefore, cause a population bias.

### 2.3. Neurodevelopmental Assessments

According to our clinical routine follow-up, all of the infants had a neurodevelopmental assessment at 24–30 months of age using the Griffiths’ Mental Development Scales [22].

The study protocol was approved by the Ethics Committee of our Institution (ID: 3419, prot. N. 0039681/20) and informed consent was obtained from parents.

### 2.4. Statistical Analysis

Data were presented as mean values and range for continuous variables normally distributed, and as count and percentages for categorical variables. The comparison between preterm children and the control group for the SDSC total as well as the 6 factor scores and age was performed using the non-parametric Mann–Whitney U test, and the comparisons of the gender were performed with Fisher’s exact test. A 2-tailed value of *p* < 0.05 was considered significant.

Subcategory analyses were performed using the previously described sleep categories.

The sample size (n = 217) was calculated considering the number of preterms (<34 weeks gestation) born in our region during the study period (N = 340), the confidence interval (n = 4), and the confidence level (95%).

## 3. Results

During the study period, 217 preterm infants (104 male, 113 female) with a mean gestational age of 27.6 weeks (range: 25–33) fulfilled the inclusion criteria. All of the infants reported a score at the Griffiths’ Mental Development Scales within the normal range. The SDSC was completed by the mothers of the preterm infants at a mean age of 18 months (range 6–36 months).

The questionnaire was also completed by the mothers of 129 typically developing children (62 male, 67 female) with a median age of 20 months (range 6–36 months). This control group presented the same age and gender distribution of the preterm group (*p* > 0.05).

### 3.1. SDSC Results in Preterm Population

An abnormal total sleep score (>70) was found in 3/217 children born prematurely (1.3%), and an abnormal score on at least one SDSC factor was found in 36 children (16%). Preterm infants were subdivided according to the birth gestational age (extremely preterm < 28 ws; very preterm < 32 ws; and moderate preterm < 34 ws) and the age of assessments (<24 months and ≥24 months). No significant differences (*p* > 0.05) on intergroup comparison were reported for SDSC total or factor scores among the 3 GA and the two ages of assessment.

### 3.2. SDSC Results in Term Born Population

An abnormal total sleep score (>70) was found in 4/129 term-born children (3%), and a total of 36 children (27%) had an abnormal score on at least one of the SDSC factors.

### 3.3. Comparison of Sleep Disorders between Preterm and Control Group

No statistically significant differences between the two groups were observed in the SDSC total or the factor scores (*p* > 0.05) (Table 1). 

## 4. Discussion

Sleep disorders constitute precursors and potential early indicators of psychopathology (e.g., regulation problems, attention problems, and aggression) at early ages [20,21,23]. In addition, in a considerable percentage of cases, sleep disturbances that occur in infancy persist during the development [23].

Therefore, an early identification of sleep disturbances in infants and toddlers appears necessary in order to ensure a helpful early intervention and to prevent the development of neurobehavioral problems and other sequalae [1,17,18,19,20].

In the present study, we analysed the prevalence of sleep disorders in a population of low-risk preterm infants between 6 and 36 months of age using the SDSC validated for this age range [20]. The main point of strength in this scale is represented by the six domains of investigation (DMS, DIS, PAR, SBD, DOES, and SHY) that fully represent the most common areas of sleep disorders in infants and toddlers, and the scale could therefore help clinicians to detect the areas that require deeper investigation [23]. The SDSC [19,20,21] has the advantages of simplicity of administration and the rapid and in-depth investigation of the main sleep problems at an early age when compared with other sleep questionnaires.

Our results reported that an abnormal total sleep score was found in the 1% of the preterm population with an abnormal score on at least one SDSC factor in the 16%. The incidence of global sleep disorders in preterm children was similar to those reported in the control group in both the SDSC total and the factor scores. These findings are consistent with data available in the literature, which currently do not report a statistically significant difference in preterm infants at the same ages using different clinical tools [17,23]. Neitman et al., in a recent study [13], did not observe a gestational age-related effect on long-term sleep behavioural data in preterm infant cohorts both at toddler and early school age (5.7 years) using the Infant Sleep Questionnaire. Caravale et al. used the SDSC in preterm toddlers (aged 24 months), and reported no differences between the preterm and term-born population on sleep patterns, such as bedtime, rise time, and nocturnal/daytime sleep durations [14].

At older ages (i.e., preschool and school age children), more specific differences are reported in the scientific literature. A previous study conducted by our research group at preschool age (3–6 years) reported some specific differences between preterm children and the control group [16]. Mainly, preterm children reported higher significant median scores than the control on SDSC total scores and in specific sleep disorders (such as difficulty in initiating and maintaining sleep, sleep-disordered breathing and sleep hyperhidrosis) [16]. Similarly, Brockmann et al. [24] showed that prepuberal preterm children reported higher scores on the SDSC’s total scores and, specifically, in the sleep-wake transition disorders, sleep-disordered breathing, and sleep hyperhidrosis subscale.

Children at pre-school age are characterized by particular changes in the quantitative and qualitative organization of sleep and the distribution of the sleep-wake rhythm [24], which could be the cause of a higher incidence of sleep disorders in this age group. Furthermore, at pre-school and school age, there is an outbreak of emotional problems such as neurodevelopmental disorders and ADHD symptoms that are very common in preterm children and are worsened by poor quality of sleep, as reported in the literature [17,25,26,27,28,29,30].

The prevalence of sleep disorders in preterm children could also be linked to different risk factors associated with a longer stay of the population of new-born babies in intensive care. Some previous studies confirm this hypothesis by several examples, such as prolonged exposure to light in intensive care units [16], alteration of circadian melatonin rhythm [16,31] intrauterine stressors [13,16], and exposure to certain drugs such as analgesics and sedatives in intensive care units [13]. Moreover, the third trimester of pregnancy, which is interrupted early in this patient population, is a crucial phase for the infant’s brain plasticity processes. Thus, due to direct action on stress-regulating systems and exposure to early stressors in an intensive environment, prematurity may be correlated with altered programming of certain brain systems, such as the hypothalamic-pituitary axis and the autonomic nervous system [32]. According to parental-based questionnaire studies, children born preterm seem to have a shorter sleep duration, longer sleep-onset latency, increased night waking, greater snoring, and prolonged sleep duration [25,26,27,28,29] that seem to last longer, even affecting adolescents and young adults. A systematic review [27] on the relationship between preterm birth and sleep in children at school age concluded that prematurity is associated with earlier bedtime and lower sleep quality (in awakenings, and more non-REM stage II sleep). Furthermore, an increase in sleep problems with decreasing gestational age in preterm infants are observed in the presence of neurodevelopmental disabilities only [16,33]. In our cohort, the presence of low-risk preterm infants only excluded preterm children with severe sleep problems due to the presence of brain lesions and neurodevelopmental disabilities, especially at lower gestational age [16]. This data could also explain the general low incidence of sleep problems in our population of infants.

SDB scores were significantly higher in pre-school and school age children born preterm compared to the controls in most of the literature [24], probably due to a history of chorioamnionitis, pharyngeal hypotonia, and adenotonsillar hypertrophia.

In our population of preterm infants, no history of pharyngeal hypotonia, chorioamnionitis, or increased adenotonsillar growth was reported, and this data could explain the presence of SBD scores similar to those of term-born infants.

Our study has several limitations that should be taken into account while interpreting study findings. The first is related to the use of a non-objective screening measure (parent-rated questionnaires) in which subjective ratings related to the parents’ perception of sleep/wake rhythm only are available. Previous studies conducted and based on a parent’s personal perception have shown how this can often vary across the country. This evidence could be based on external factors, such as geographic origin, child’s lifestyle, and personal concept of sleep and night rest and other health behaviors (e.g., diet, exercise, etc.) [34]. More objective methodologies could eliminate such bias related to personal perception.

Another limitation could be related to the absence of information on potential neonatal risk factors (e.g., BMI, breastfeeding, use of caffeine in the mother, length of hospitalisation) that were not systematically collected but that have been found to have a potential effect on the mechanisms underlying sleep disorders [33].

Furthermore, the specific family and environmental setting, such as the presence of siblings with or without chronic medical problems, has not been investigated. This data could be of interest as this could change the lifestyle and the personal perception of the parent interviewed.

At last, the control group is from a nursery setting in “normal” controls, and the responses of the mothers might be far more casual in this setting than in a hospital Neurology follow-up clinic, where mothers would be more likely to be very diligent and under stress.

## 5. Conclusions

In conclusion, our results demonstrated that low-risk preterm infants and toddlers show a similar incidence of sleep disorders to their term-born peers. 

However, the literature has confirmed that this incidence may be significantly higher during the school-age period (24–28). An interesting perspective for the future could be related to the follow-up of these same patients studied from 6 to 36 months, re-evaluating sleep patterns by administering the same questionnaire during school age.

## Figures and Tables

**Table 1 jpm-13-01091-t001:** SDSC scores in preterm children and in the control group.

	PAR	DIS	DMS	DOES	SHY	SBD	T Sccore Totali
Preterm group	47.4 ± 9	46.5 ± 9.12	50.3 ± 9.5	47.5 ± 8.3	49.15 ± 8.9	46.6 ± 7.22	46.3 ± 8.2
Control group	49 ± 11.1	49 ± 12.1	49 ± 11.3	49 ± 11.6	49.2 ± 10	49 ± 9.2	49 ± 10

(DIS) Disorders in initiating sleep; (DMS) difficult in maintaining sleep; (SBD) sleep-disordered breathing; (PAR) parasomnias; (DOES) disorders of excessive somnolence factor; (SHY) sleep hyperhidrosis.

## Data Availability

Not applicable.

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
