# Peer review of "Sleep Disorders in Low-Risk Preterm Infants and Toddlers"

_jpm, 2023, doi:10.3390/jpm13071091_

Round 1
Reviewer 1 Report
General Comments:
The authors compared SDSC results as scored by parent caregivers between 217 preterm toddlers with no significant neurologic sequelae from their NICU stay with those from 129 typically developing term toddlers and report no differences in total or subset scores between the 2 groups. The study is valuable since sequelae of NICU stays and preterm born children in particular may be significant and understanding issues early is important to facilitate appropriate screening and/or interventions. That said, I have several issues with the study as presented, not design so much as the volume of extraneous information presented that really does not pertain or relate to the specific findings of the study. The manuscript could be substantially shortened by removal of these areas (see specific comments for suggestions). Also, I think a major flaw in the discussion is the lack of discussion of how/why the current data differ from other reported data. There are far too many references which would be remedied by removing a lot of the extraneous sections alluded to above in the Intro and Discussion.
The manuscript will require moderate copyediting for diction and grammar
Specific Comments:
Abstract:
- There is a lack of structure to the abstract, specifically it could flesh out a bit more the subject population, the specifics of the sleep disorders reported vs those tested for as well as if there was a specific hypothesis, for example –‘we postulated that there would be differences in xxx factors of the SDSC between preterm and term children” or something like that. As a reader I typically skim abstracts to determine if I want to read the entire manuscript and this would not give me adequate information to determine if I would go on and read the paper.
Introduction:
- Sleep can be assessed either using scoring tools or via formal polysomnography – it would be helpful for the authors to indicate which studies referenced used 1 which (or both possibly) of these methods as each has limitations.
- The detailed description of the sleep states found in infants of different gestational ages is probably better placed in the Discussion as the introduction, as it currently is, is very long.
- Similar to the comment above, the intro takes a very long time to get to the point of the study, the specific question(s)/aim(s) being evaluated and the hypothesis(es) being evaluated. Please reduce the length to make if more focused on getting to this point.
Methods:
- Was the SDSC only completed by 1 caregiver (sounds like primarily the mother)? If so, why were impressions from both caregivers (if present) taken into account. Also, can you comment on if the survey can be influenced by caregivers who have other children (and can therefore compare the subject to his/her siblings) versus caregivers of a single child where these possible biases would be at least fewer? Same question for both cases and controls would apply.
- Can you please expand on how the subcategories of sleep were evaluated statistically? (which ones were “combined” into categories etc and, if that was based on the categories outlines earlier in the methods just a statement that subcategory analyses were performed using the previously described categories would be sufficient
- I do not see a power analysis to determine if the sample size would be adequate to detect whatever the authors determined would be a clinically meaningful difference in outcomes.
Results:
- This section seems very sparse in the prose section (though the Table contains the main pertinent information) but, related to the comment above, I would think it would be valuable to know how many of the cases and controls had other siblings (with or without chronic medical problems).
- There are no p values or odds ratios presented in the results or in the Table to support the lack of difference between the 2 groups and they need to be.
Discussion:
- Paragraph 3 – in the introduction it is stated that 10-30% of children experience sleep problems whereas the current study reports only a 1% incidence and that this is typical of current literature. These statements are discordant in the reported percentages of sleep problems and needs further expansion/reconciliation. If I have misinterpreted this, the manuscript should be clarified to better indicate that these reported percentages are NOT different from what is stated in the intro to avoid confusion by other readers.
- There is a fair amount of discussion summarizing the current literature, including quoting some results that appear different from the current stud findings but very little expansion on why these differences might be present, wither due to patient factors or study design differences. Such discussion needs to be significantly added onto.
- Page 4 of the Discussion suggests that there are numerous NICU factors which could contribute to sleep disorders but does not go on to explain why the incidence of sleep disorders in the study population did not appear to reflect this (since there were no differences between cases and controls – this needs further discussion.
- While interesting, since it was not assessed for very specifically, the discussion section regarding SDB and academic abilities could be removed – it just is not really pertinent to the findings of the current study.
- In fact, as I read more, there are several sections in the discussion not really related to the current study aims and could be deleted – the discussion reads more like a review article on sleep problems in preterm infants rather than an expansion on the findings of the current study.
- The influence of other home environment factors on scoring the SDSC as a limitation is also not really comment on and should be.
Conclusions:
- I’m a bit confused by the message given in the latter paragraphs of this section. The results show NO DIFFERENCE in sleep scores between preterm and term toddlers BUT the conclusions go on to indicate that preterm infants should be carefully screened so sleep disorders can be detected. This suggest that the authors, despite the results, still believe that preterm infants are at higher risk than term infants which could be confusing to the reader. The comments about creating a healthy sleep environment, while important, also are not supported (because they are not a part of the study design) by the results and should be removed
The manuscript will require moderate copyediting for diction and grammar
Author Response
"Please see the attachment

Reviewer 2 Report
I congratulate the authors on the conduct of a study regarding a very relevant topic
There are some significant suggestions:
1. There seems to be no sample size calculation done - this would be essential to power the study to detect differences. otherwise we can only conclude that "with this convenience Samp;ing (cross sectional data over a period of time), these were our findings” we cannot draw conclusions that there is no difference in sleep issues between 2 groups
2. The introduction and the discussion are far too long and switching from one topic to the other
for eg: problems in sleep during NICU care are a very different set of problems ; and need not be discussed in so much detail. the study is about problems later on in infancy and early childhood
the flow of the paper is therefore much disturbed
3. the sleep patterns in infants (6 months old) are affected largely by the family’s structure and sleep hygiene practices, breast feeding practices etc.. The patterns in 36 month children are on the contrary affected by a large number of factors -- even screen time, diet lets.
although the authors have highlighted some of these points in the discussion, I am curious to know why the authors chose such a wide AGE - range for assessment (6 to 36 months) ?
4. The control group is from a nursery setting in “normal” controls, the mothers’ responses may be far more casual in this setting than in a hospital Neurology follow up clinic where mothers would be very diligent and under stress.
this is a very significant bias in reporting the primary outcome of this study
5. there are several English language edits that need to be made:
for eg; in introduction -- “spy” of a different maturation ???
6. the abstract is not in any IMRAD format-- please comply with the manuscript guidelines of the journal as suitable
English needs some editing as described above
Author Response
"Please see the attachment
